# The Impact of the COVID-19 Pandemic on Diabetes Self-Management in Saudi Arabia

**DOI:** 10.3390/healthcare12050521

**Published:** 2024-02-22

**Authors:** Ibrahim Sales, Ghada Bawazeer, Ahmad Abdul-Wahhab Shahba, Hadeel Alkofide

**Affiliations:** 1Department of Clinical Pharmacy, College of Pharmacy, King Saud University, Riyadh 11451, Saudi Arabia; gbawazeer@ksu.edu.sa (G.B.); halkofide@ksu.edu.sa (H.A.); 2Department of Pharmaceutics, College of Pharmacy, King Saud University, Riyadh 11451, Saudi Arabia; shahba@ksu.edu.sa

**Keywords:** COVID-19 pandemic, diabetes, self-management, Saudi Arabia, questionnaire

## Abstract

The COVID-19 pandemic disrupted healthcare worldwide, potentially impacting disease management. The objective of this study was to assess the self-management behaviors of Saudi patients with diabetes during and after the COVID pandemic period using the Arabic version of the Diabetes Self-Management Questionnaire (DSMQ). A cross-sectional study was conducted in patients aged ≥18 years diagnosed with type 2 diabetes mellitus who had at least one ambulatory clinic visit in each of the specified time frames (Pre-COVID-19: 1 January 2019–21 March 2020; COVID-19 Time frame: 22 March 2020 to 30 April 2021) utilizing the DSMQ questionnaire, with an additional three questions specifically related to their diabetes care during the COVID pandemic. A total of 341 patients participated in the study. The study results revealed that the surveyed patients showed moderately high self-care activities post-COVID-19. Total DSMQ scores were significantly higher in patients aged >60 years versus younger groups (*p* < 0.05). Scores were significantly lower in patients diagnosed for 1–5 years versus longer durations (*p* < 0.05). Patients on insulin had higher glucose management sub-scores than oral medication users (*p* < 0.05). Overall, DSMQ scores were higher than the pre-pandemic Saudi population and Turkish post-pandemic findings. DSMQ results suggest that, while COVID-19 negatively impacted some self-management domains, the Saudi patients surveyed in this study upheld relatively good diabetes control during the pandemic. Further research is warranted on specific barriers to optimize diabetes care during public health crises.

## 1. Introduction

The COVID-19 pandemic had a diverse range of effects on the self-management of diabetes. Many aspects of diabetes management were disturbed during the pandemic. The interruption of healthcare services due to the implementation of lockdown measures, the burden on healthcare systems due to the high mortality of COVID-19, and the fear of contracting the COVID-19 infection significantly affected the provision of diabetes care [1,2]. As a result, there was a decline not only in routine ambulatory visits, but also in non-COVID-related preventive and emergency care [3,4,5,6]. The restriction on the various lifestyle routines and daily activities such as diet and physical activity, as well as self-monitoring, timely access to medications and supplies, and healthcare utilization-seeking behaviors, were directly affected by the pandemic [7]. In addition, stress, anxiety, and depression were highly prevalent during the pandemic and may have indirectly influenced the mental well-being of individuals and coping patterns [7,8,9,10].

In the Kingdom of Saudi Arabia (KSA), several studies have explored the impact of the COVID-19 pandemic on the activities and behaviors of patients with diabetes in a variety of ways. Fatani et al. reported that the lockdown had detrimental effects on physical activity and sleeping hours, but minimal effect on eating habits, which may have mitigated the negative effect on glycemic control parameters compared to the pre-COVID period [11]. Aldaghri et al. also reported that dietary habits significantly changed in content, number, and mealtimes compared to a control group [12]. A cross-sectional study in 394 individuals with diabetes in Jeddah Province showed that compliance with diet and physical activity was reduced in the post-lockdown period compared to the pre-lockdown period (25.1% vs. 27.7% and 31% vs. 35%, respectively) [13]. In addition, among the same population, compliance with medications and regular self-testing of blood glucose were reported in 88.3% and 46.2% of individuals, respectively.

Even in the era predating COVID-19, patients with diabetes struggled with self-management practices such as regular monitoring of blood glucose levels, medication adherence, balanced diet, exercise, and health checkups [14,15]. Almiqbal et al. investigated the association between glycemic control and diabetes self-management, depression, and health literacy in a sample of patients with type 2 diabetes mellitus in KSA [16]. The study found that scores on the Diabetes Self-Management Questionnaire (DSMQ) were low for healthcare use, especially adherence to medical appointments. In addition, low scores were reported in the physical activity subscale. Another study by Alqahtani found a high prevalence of inadequate management of all the DSMQ subscales in a population in Najran city in the southern region of KSA [17].

Effective diabetes self-management practices empirically require a positive patient attitude, a supportive environment, and a well-established relationship with a healthcare provider. These critical components were considerably disrupted during the COVID-19 pandemic. Although numerous measures, including telemedicine, were quickly introduced after the lockdown, the effects of such dramatic shifts and fluctuations in care delivery on patients’ abilities to engage in diabetes self-management is worthy of investigation. We sought to evaluate the impact of COVID-19 on a demographically comparable population studied by Almiqbal et al. [16]. The objective of this study was to assess the self-management behaviors of Saudi patients with diabetes during and after the COVID pandemic period using the Arabic version of the DSMQ.

## 2. Materials and Methods

### 2.1. Study Design and Setting

This was a cross-sectional study conducted at King Saud University Medical City (KSUMC) in Riyadh, KSA, from January 2022 to August 2023.

### 2.2. Study Population and Sampling

This study was embedded in a cross-sectional study that examined the impact of COVID on diabetes control. The main study included 1777 patients who met the following inclusion criteria: ≥18 years of age, patients with type 2 diabetes, and those who had at least one healthcare encounter in KSUMC ambulatory care clinics during the two pre-specified time periods—pre-COVID-19 (between 1 January 2019 and 21 March 2020) AND during/post-COVID-19 (between 22 March 2020 and 30 April 2021). The study was conducted from January 2022 to August 2023.

The sample size was calculated using RASOFT calculator, assuming a margin of error of 5%, a confidence level of 95%, a 50% response, and a population size of 1777. The calculated sample was 317. Patients for the DSMQ survey were randomly selected from the identified population (using Microsoft Excel ^®^ random generation sequence (Microsoft Corporation, Redmond, United states)) and were contacted through the hospital call center or via clinic phone. We originally planned to distribute the questionnaire, but, due to logistics in identifying the population during their hospital visits and the challenges in obtaining responses using digital media, we amended our IRB. We conducted the questionnaire using the hospital call center.

### 2.3. Data Collection and Instruments

The DSMQ is a validated instrument developed by Schmitt et al. from the Research Institute at the Diabetes Academy Mergentheim [18]. The survey was previously administered in Arabic to a demographically comparable population from the same healthcare facility (KSUMC) [16]. The questionnaire is a 16-item instrument developed to investigate the relationship between self-management in patients with type 1 and type 2 diabetes and glycemic control [18]. It contains four domains covering aspects related to dietary control, blood glucose management and medication adherence, physical activity, and physician contact/appointments. It has been validated, translated into many languages (including Arabic), and used in the Gulf region (Saudi Arabia, Kuwait, and Oman) [16,17,19,20]. The DSMQ has been utilized extensively in various settings, contexts, and countries such as Egypt, Ethiopia, Hungary, India, Indonesia, Iran, Kuwait, Nigeria, Philippines, Qatar, Romania, Spain, Thailand, and the United States of America [19,21,22,23,24,25,26,27,28,29,30,31,32,33]. Some of the highest scores have been published in the German (7.8) and Chinese (7.79) populations, and lower scores have been reported in the United Kingdom (4.14) and Pakistan (3.96) [8,34,35,36]. Permission to use the questionnaire was obtained through Mapi Research Trust (https://eprovide.mapi-trust.org (accessed on 9 March 2021)). In the current study, three additional items on overall COVID-19 impact upon diabetes control, diet, and exercise were asked, to end up with a total of 19 questions. The scale includes five subscales: glucose management (1, 4, 6, 10, 12), diet control (2, 5, 9,13), physical activity (8, 11, 15), use of health services (3, 7, 14), and overall COVID-19 impact (17,18,19). Item 16 was not included in any subscales. A Likert scale consisting of 4 items was used, and each item on the scale was scored from 0 to 3 (0—does not apply to me, 1—applies to me a little, 2—applies considerably to me, 3—applies very much to me). Items 5, 7, 10, 11, 12, 13, 14, 15, 16, 17, 18, and 19 on the scale were scored in reverse. For calculating the total DSMQ score and subscales, the relevant item scores were summed then transformed to a scale ranging from 0 to 10 (summed actual score/summed theoretical score × 10). For example, the subscale “Dietary Control” involves 4 items, so the sum theoretical score = 12; therefore, a summed score of 9 equals to a transformed score of 9/12 × 10 = 7.5. A score approaching 10 indicated greater diabetes self-management and scores ≤ 6 were considered to be reflective of poor self-management. The reliability (internal consistency) of the surveys was assessed through Cronbach’s alpha value [37]. Duplicate data entries were removed and data about the patients’ comorbidities, age, sex, diabetes duration, and medications were verified from the patients’ medical records when needed.

### 2.4. Data Analysis

#### 2.4.1. Software

The data analysis was primarily conducted using the Python programming language (version 3.9.13) within a Jupyter Notebook environment (jupyter_core: 4.11.1, notebook server: 6.4.12). A variety of packages, including pandas, numpy, seaborn, matplotlib, itertools, and statannotations, were employed for tasks such as presenting data, organizing them into groups, validating them, manipulating data frames, and generating visualizations. Parts of this manuscript were crafted with the assistance of automated writing programs like Claude (Anthropic, San Francisco, CA, USA) operated by POE (Quora, Mountain View, CA, USA), Copy.ai platform (Copy.ai, Memphis, TN, USA), as well as Microsoft’s AI chatbot (Microsoft, Redmond, WA, USA). Some Python code scripts used for associated data processing benefited to some extent from guidance provided by those artificial intelligence tools. Notwithstanding, the authors hold accountability for the concepts explored, the substance covered, and the final form of the manuscript.

#### 2.4.2. Statistical Analysis

The normality of the data was determined using the Shapiro–Wilk test in the Scipy.stats Python package [38]. For dependent variables that were binary (e.g., pass/fail), the data were analyzed statistically using a chi-square test for independence in a contingency table to evaluate two independent samples. When there were more than two samples, a Bonferroni post-hoc adjustment following chi-square was utilized. These analyses were conducted using functions in the Scipy Python package (version 1.9.1) [39,40].

Other dependent variables that were discrete or continuous were assessed using the Mann–Whitney U test for two independent samples (statannotations Python package, version 0.5.0) or the Wilcoxon signed-rank test for two paired samples (Scipy.stats Python package, version 1.9.1). For more than two samples, a Kruskal–Wallis H test followed by post-hoc Dunn’s test with Bonferroni correction was applied, drawing on functions from the Pingouin (version 0.5.3) and Scikit-Posthocs packages (version 0.7.0) [41].

Correlational analysis involved Spearman’s correlation test using Scipy functions (version 1.9.1) [42]. For all statistical tests, a *p*-value of ≤ 0.05 was considered statistically significant.

### 2.5. Ethical Considerations

This study was approved by the King Saud University College of Medicine Institutional Review Board (No. E-21-5883). Participants were informed that their participation was voluntary, and they had the right to decline completing the questionnaire at any point in time.

## 3. Results

### 3.1. Study Overview

The study involved 341 participants who were enrolled in the DSMQ study. The demographic data showed a balanced distribution of sex, where 60% of patients were female, while the majority of patients (97%) were Saudi. For age, 43% were 61–70, 30% were 51–60, and 17% were older than 70 years old. For the medications, 49% were taking oral medications only, 38% were taking oral medications as well as insulin, and the remaining 13% were taking insulin only. The average score for each item ranged from 1.76 to 2.83 (out of 3), which indicates good diabetes self-management during COVID-19. The survey consistency study indicated that the majority of the survey dimensions had acceptable (>0.5) Cronbach’s alpha values. However, only the dimension of healthcare use, which involved three items, showed a low Cronbach’s alpha value (0.26). This agrees with what Schmitt reported regarding healthcare use, showing a marginal consistency value [18]. However, the DSMQ sum scale based on 16 or 19 items showed high Cronbach’s alpha values (>0.7), which reflects good overall internal consistency of the survey items together.

### 3.2. DSMQ Findings of the Current Study

The DSMQ study results indicated that Saudi adults had moderate levels of diabetes self-care management and behaviors after the COVID-19 pandemic based on their DSMQ scores (Table 1). The mean DSMQ sum scale score for the 16 core items was 7.29 ± 1.4, indicating moderately high self-care activities overall. Scores were highest for healthcare use (8.6 ± 1.77), followed by overall self-care rating (8.45 ± 2.65), dietary control (6.82 ± 2.17), and physical activity (6.12 ± 2.95). The overall COVID-19 impact score was 7.81 ± 2.37, but it is important to note that these three questions were reverse-scored, so a higher score indicates lower perceived pandemic impact. Notably, after adding the additional reversed pandemic questions, the 19-item score slightly increased to 7.37 ± 1.28, compared to the original 16-item score.

### 3.3. Influence of Patient Demographics on DSMQ Findings

Participant sex showed no significant effect on any DSMQ subscale nor on the sum scale; however, patient age had a significant (*p* < 0.05) effect on diet control, overall rating of self-care, and DSMQ sum scale (Table 2). In particular, patients aged >60 years old showed significantly higher (*p* < 0.05) ratings of diet control and overall self-care. Although all other subscales showed no significant differences, the DSMQ sum scale showed significantly higher scores in elderly patients (Table 2). Regarding the time since diagnosis of diabetes, patients that were diagnosed with diabetes 1–5 years ago showed significantly (*p* < 0.05) lower glucose management scores compared to corresponding patient groups with longer diabetes history (Table 2 and Table 3). However, diabetes duration showed no significant effect on all other subscales nor on the sum scale. Similar findings were observed for the effect of medication type, where patients who were on insulin therapy (either alone or in combination) showed significantly (*p* < 0.05) higher glucose management scores compared to corresponding patients who used oral medications only (Table 2 and Table 3). No other significant effects were observed for medication types in any of the other subscales nor for the sum scale. It is worth mentioning that this study involved a very low percentage of non-Saudi participants (3%) and the statistical analysis showed that no significant differences were found between the Saudi and non-Saudi population within the DSMQ sum scale nor the subscales.

## 4. Discussion

This study focused on the impact of the COVID-19 pandemic on diabetes self-management in the Saudi Arabian population using the DSMQ. It is worth mentioning that the DSMQ was conducted in the same hospital system with a similar patient population prior to COVID-19 by Almiqbal et al. [16]. However, our study included specific questions regarding the impact of the COVID-19 pandemic on diabetes control. Interestingly, the current study showed a relatively lower glucose management score compared to the Saudi population before COVID-19; however, there were higher subscale scores for diet control, physical activity, and, particularly, healthcare use that showed about a 1.8-fold increment compared to the pre-COVID-19 DSMQ (Table 4) [16]. Furthermore, the current study showed higher DSMQ total scores, as well as scores for all DSMQ subscales, compared to post-COVID-19 DSMQ findings in the Turkish population (Table 4) [43].

When compared to DSMQ results prior to COVID-19 as well as the results in the Turkish population, patients in this study generally had better DSMQ sum scale scores [10,16,17,43,44,45,46]. Previous studies suggested that the COVID-19 pandemic had a significant impact on diabetes self-management in the KSA population. The challenges faced by individuals with diabetes include reduced compliance with medical treatment, lifestyle changes, increased psychological distress, and limited access to healthcare services; however, the findings of this study have several important implications related to diabetes care in the post-COVID-19 era [10,11,12,13,47,48].

In the current study, participants reported high glucose management scores. Two studies in similar patient populations to this study had slightly higher scores in this domain and one had lower scores, but it appears as the COVID-19 pandemic did not lead to significant changes in glycemic management [16,45,46]. This was directly expressed by the study participants in their answers to the three COVID-19 questions. It is interesting to note that the glucose management scores in this study were substantially higher than those reported in the southern region of KSA, the scores of Saudi Arabian healthcare professionals, and post-COVID-19 scores in Turkey [17,43,44]. Difficulties in glucose management may be due to common barriers reported in both local and international studies, which have mentioned that adherence and access to healthcare services was decreased during the pandemic. A cross-sectional study conducted in Jazan, KSA, involving 394 patients found that the COVID-19 lockdown significantly reduced the levels of compliance, medical treatment, and lifestyle habits among Saudi patients with diabetes [49]. A one-year follow-up study examined the impact of the COVID-19 pandemic on diabetes self-management globally. While not specific to the KSA population, the study highlighted the challenges faced by individuals with diabetes during the pandemic, including disruptions in healthcare access, changes in lifestyle habits, and increased psychological distress [48]. Difficulties accessing medications, a vital component of therapy, due to lockdown restrictions in movement or medication shortages may have been the primary cause as opposed to patient negligence.

There were several positive results that show promise for the future care of patients with diabetes in KSA. When compared to pre-COVID-19 results by Almigbal et al., participants reported higher scores for dietary control and physical activity [16]. Furthermore, healthcare use approximately doubled post-COVID-19. Dietary control, physical activity, and healthcare system utilization are highly emphasized core components of Diabetes Self-Management Education and Support (DSMES) [50]. Furthermore, lifestyle changes, including diet and exercise, can lead to a reduction in hemoglobin A1C of up to 2% [51,52]. The dietary control scores reported in this study were the highest reported in KSA and higher than post-COVID-19 scores in Turkey [16,17,43,45,46]. The only exception was the dietary scores reported in healthcare providers; however, this population was different than our study and it is expected that healthcare professionals dealing with patients with diabetes would have better dietary control, due to more knowledge about the dietary recommendations and to access to healthier food choices [44]. Likewise, physical activity scores were amongst the highest reported in KSA to date. During the pandemic, patients may have dedicated more time towards exercise and been more conscious of their dietary habits. This may have been the case for many patients due to reduced opportunities for gathering with family and friends, which may have otherwise made adherence to dietary changes more difficult due to societal or peer pressure. It is also possible that patients were more encouraged to make lifestyle modifications due to the increased risk of poor COVID-19 outcomes and mortality in patients with diabetes.

Furthermore, the healthcare use scores were by far the most impressive scores reported in both KSA and Turkey post-pandemic [10,16,17,43,44,45,46]. Improved healthcare utilization is an essential component of diabetes care in general, and a crucial component of the Saudi Vision 2030 [53]. One of the national healthcare transformation objectives includes facilitating access to healthcare services, and an important related initiative is the development of a healthcare model that places more emphasis on the prevention of diseases rather than treatment [54]. Telehealth is a means of expanding patient access, and many of the participants in this study increased their healthcare utilization via virtual medical encounters. The Ambulatory Care Practice Research Network writing task force of the Saudi Society of Clinical Pharmacy has called for an expansion of virtual clinics and Telehealth to achieve this objective [55].

While overall self-care was upheld, some disparities emerged across demographic subgroups of Saudis with diabetes. Newly diagnosed patients reported lower DSMQ scores compared to those with longer diabetes duration, highlighting the need for more intensive education and skills training early in the disease course. This is especially important during crises when usual health services are interrupted. The higher scores among elderly Saudis may reflect greater health consciousness with age or concerns about diabetes complications. Although not reaching a level of statistical significance, Alkhormi et al. also reported that patients ≥65 accounted for a greater percentage of participants with appropriate self-management of diabetes during COVID-19 as opposed to their younger counterparts [10]. This differs, however, from the post-COVID-19 results in Turkey and pre-COVID results from Najran, which indicated that patients who had diabetes for a longer duration or were elderly had poorer self-management [17,43]. This may indicate that the patients in this study were better educated regarding diabetes care. Finally, those using insulin had better glucose monitoring skills than oral medication users. This was possibly due to the need for frequent self-injections.

There are some limitations that should be mentioned. First, the study included a small sample size from one hospital system. Second, bias due to over- or under-reporting cannot be ruled out when utilizing a self-reported questionnaire. Third, recall bias may have occurred in some participants who participated in the later stages of the survey. Finally, this manuscript did not include clinical data such as the hemoglobin A1C or other related patient vitals.

## 5. Conclusions

In conclusion, this cross-sectional study provides valuable insights into the maintenance of diabetes self-management among Saudis during the significant lifestyle and healthcare disruptions imposed by COVID-19. The DSMQ results suggest effective adaptation by Saudi patients, but also reveal target groups that require more self-care support, particularly during public health emergencies. The healthcare transformation sector of the Saudi Vision 2030 should prioritize these vulnerable populations that require more care and attention when developing and implementing future policy changes.

## Figures and Tables

**Table 1 healthcare-12-00521-t001:** Descriptive analysis of total score and subscales of DSMQ (n = 341).

Parameters	Score (Mean ± SD) n = 341
Glucose management subscale	7.35 ± 1.94
Dietary control subscale	6.82 ± 2.17
Physical activity subscale	6.12 ± 2.95
Healthcare use subscale	8.6 ± 1.77
Overall rating of self-care	8.45 ± 2.65
DSMQ sum scale (Q1–Q16)	7.29 ± 1.4
COVID-19 impact subscale	7.81 ± 2.37
DSMQ and COVID-19 Impact sum scale (Q1–Q19)	7.37 ± 1.28

**Table 2 healthcare-12-00521-t002:** Influence of patient demographics and clinical features on DSMQ domain scores (*n* = 341).

Characteristics		Glucose Management Subscale		Dietary Control Subscale		Physcial Activity Subscale		Healthcare Use Subscale		Overall Rating of Self-Care		COVID-19 Impact Subscale		DSMQ Total (Q1–Q19)	
	n	Mean ± SD	*p*-Value	Mean ± SD	*p*-Value	Mean ± SD	*p*-Value	Mean ± SD	*p*-Value	Mean ± SD	*p*-Value	Mean ± SD	*p*-Value	Mean ± SD	*p*-Value
**Sex**															
Female	206	7.2 ± 2.0	0.18 ^a^	6.9 ± 2.1	0.68 ^a^	5.9 ± 2.8	0.07 ^a^	8.57 ± 1.7	0.29 ^a^	8.3 ± 2.8	0.14 ^a^	7.7 ± 2.4	0.62 ^a^	7.3 ± 1.3	0.13 ^a^
Male	135	7.5 ± 1.9		6.8 ± 2.3		6.4 ± 3.1		8.65 ± 1.88		8.7 ± 2.4		7.9 ± 2.3		7.5 ± 1.3	
**Age**															
18–60	138	7.1 ± 2.0	0.07 ^a^	6.4 ± 2.3	0.02 ^a^	6.3 ± 3.0	0.42 ^a^	8.52 ± 1.83	0.49 ^a^	8.2 ± 2.6	0.02 ^a^	7.6 ± 2.4	0.07 ^a^	7.2 ± 1.4	0.02 ^a^
>60	203	7.5 ± 1.9		7.1 ± 2.0		6 ± 2.9		8.65 ± 1.73		8.6 ± 2.7		8 ± 2.3		7.5 ± 1.2	
**Duration_of_Diabetes**															
1–5 years	25	6.2 ± 1.8	0.005 ^b^	6.8 ± 2.0	0.94 ^b^	5.8 ± 2.7	0.13 ^b^	8.8 ± 1.36	0.56 ^b^	7.9 ± 2.9	0.35 ^b^	8.1 ± 2.3	0.87 ^b^	7.1 ± 1.1	0.25 ^b^
6–11 years	61	7.3 ± 2.0		6.9 ± 2.2		6.8 ± 2.7		8.4 ± 1.89		8.4 ± 2.6		7.8 ± 2.2		7.5 ± 1.4	
12 years or more	246	7.5 ± 1.9		6.8 ± 2.2		6.1 ± 3.0		8.66 ± 1.74		8.5 ± 2.6		7.8 ± 2.4		7.4 ± 1.3	
**Type of anti-diabetic** **medications**															
Insulin	45	7.8 ± 1.8	0.0003 ^b^	6.6 ± 1.9	0.43 ^b^	6.2 ± 3.2	0.12 ^b^	8.91 ± 1.6	0.34 ^b^	8.4 ± 2.9	0.94 ^b^	7.4 ± 2.4	0.27 ^b^	7.4 ± 1.3	0.89 ^b^
Oral medications	166	6.9 ± 1.9		7 ± 2.1		6.4 ± 3.0		8.56 ± 1.71		8.4 ± 2.6		7.9 ± 2.4		7.3 ± 1.3	
Oral medications + Insulin	130	7.7 ± 1.9		6.6 ± 2.3		5.8 ± 2.7		8.54 ± 1.9		8.5 ± 2.6		7.9 ± 2.3		7.4 ± 1.3	

**^a^** Mann–Whitney U test; **^b^** Kruskal–Wallis test. Significant *p*-values (<0.05) are highlighted by red color.

**Table 3 healthcare-12-00521-t003:** Pairwise comparisons of significantly different findings between groups.

Characteristics	Pairwise Comparision between Groups	*p*-Value
**Duration** **of diabetes**	1–5 years vs. 6–11 years	0.03 ^c^
	6–11 years vs. 12 years or more	1.00 ^c^
	1–5 years vs. 12 years or more	0.004 ^c^
**Type of anti-diabetic** **medications**	Insulin vs. Oral medications	0.01 ^c^
	Oral medications vs. Oral medications + Insulin	0.001 ^c^
	Insulin vs. Oral medications + Insulin	1.00 ^c^

**^c^** Dunn post-hoc test with Bonferroni correction. Significant *p*-values (<0.05) are highlighted by red color.

**Table 4 healthcare-12-00521-t004:** Comparative analysis of total score and subscales of the current DSMQ study along with other relevant studies.

Parameters	DMSQ Score		
	Turkish Population Post-COVID-19 (Adapted from [43]) n = 378	Saudi Population Post-COVID-19 (Current Study) n = 341	Saudi Population before COVID-19 (Adapted from [16]) n = 352
Glucose management	5.18 ± 0.24	7.35 ± 1.94	7.8 ± 2.3
Dietary control	5.20 ± 1.15	6.82 ± 2.17	6.5 ± 1.5
Physical activity	5.10 ± 0.22	6.12 ± 2.95	5.8 ± 1.1
Healthcare use	5.24 ± 0.56	8.6 ± 1.77	4.8 ± 1.2
DSMQ sum scale (16 items)	5.25 ± 1.04	7.29 ± 1.4	NA

## Data Availability

The data presented in this study are available on request from the corresponding author.

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
