# Peer review of "The Impact of the COVID-19 Pandemic on Diabetes Self-Management in Saudi Arabia"

_healthcare, 2024, doi:10.3390/healthcare12050521_

Round 1

Reviewer 1 Report

Comments and Suggestions for Authors

The Impact of the COVID-19 Pandemic on Diabetes Self- management in Saudi Arabia

Thank you for considering me as a potential reviewer for this paper. Given the topic's popularity during the COVID-19 pandemic, I have some concerns regarding this study. I am not convinced about the importance of the study to this journal's readers. In particular, I am concerned about the originality of this research as there are many research published on the same topic in the same country. Here are some suggestions to improve the quality of the paper.

Title:

It does not reflect the study setting as the research was conducted in one centre, King Saud University Medical City in KSA, and for sure, it will not represent the whole country.

Abstract:

The study objectives mentioned in the abstract section differ from those mentioned at the end of the Introduction section.

Introduction:

-Each study is developed based on research questions or hypotheses; the authors should indicate their research questions in the introduction.

- In this section, we need to know if the paper provides new insight into the field of Diabetes Self-Management, given that there are plenty of other research papers on the topic.

-Much of the introduction (lines 52-60) is devoted to a well-known questionnaire (The Diabetes Self-Management Questionnaire); I think what is mentioned in the methods section is enough.

Methods:

"This section requires improvement, specifically in the following areas."

- It needs to be made clear when exactly the data was collected from the patients. It is mentioned that the study period is December 2021 to August 2023, which is a very long duration and how to interview patients during 2023 about events in 2021. This should be clearer.

- Patients were contacted via phone; the mechanism is not clear. Does the study team use their phone number in the medical files? If so, do patients agree to use their phone numbers for research purposes, and do the ethics regulations permit such procedures?

- The sampling procedure needs to be improved; what is the random sampling method used for selecting the study participants? Further, it is mentioned that the sampling is calculated to be 341; on what basis?, please clarify the different Parmenter used for sample size calculation. 

- It is better to move the questionnaire (DSMQ) to the appendix or supplementary material section.  

- In the methods section, the authors described the participants' demographics; this is unsuitable for this section and would be better summarized in the results section with other patient’s background characteristics, which were not mentioned.  

Results:

I strongly recommend rewriting for the following reasons:

- This section should be made shorter and crisper. Tables can be used and presented to make it easier to understand. There are more than 20 graphs, which is very difficult to follow.

- In this section, authors compared their findings with relevant previous studies; please focus on your findings only. Such comparisons can be discussed in the discussion section.

Discussion and Conclusion:

Finally, the Conclusion section should be more reflective of the policy implications of the findings.

Author Response

Thank you for taking the time to review our manuscript. Please find our responses attached. 

Reviewer 2 Report

Comments and Suggestions for Authors

With a cross-sectional study, impact is possible؟؟

Rewrite the abstract based on the strobe checklist, https://www.strobe-statement.org/checklists/ conference abstracts. Line 12 the aim of study is needed.

The methodology and results are not clear in the abstract.

Check the keywords based on the mesh.

Line 25, what’s ERI??

This doi: 10.1007/s12029-021-00679-x. And doi: 10.1007/s12029-021-00752-5. can be useful to improve the introduction. In introduction, gap and purpose should be expressed in better sentence.

Rewrite the method based on the standard checklist, the sequence of the contents is not appropriate.

Line 69 all medical records or Patients?

randomly selected, how??

 and contacted via telephone and/or Whatsapp, Convenience sampling?  telephone and/or Whatsapp  are completely different and not comparable.

Validity and reliability of the questionnaire are not mentioned.

Diabetes Self-Management Questionnaire (DSMQ).  Is needed??

 Study design. Setting, Participants, sample size calculation, Variables, Data sources/ measurement, must be clear and complete.

Of 508, 342 patients were included in the subsequent 114 score analysis. This response rate is not suitable.

Line 104 – 124 is related to the results. And the figures are not suitable.

Statistical tests are not clear.

Table 2 and 4 is unclear, also Figure 2.

There are many figures that are confusing.

Limitations and Conclusions should be more concise

Author Response

(The authors gave the same response as above.)

Reviewer 3 Report

Comments and Suggestions for Authors

Thank you for the opportunity to review this manuscript. The authors have aimed to assess the self-management behaviors of Saudi patients with diabetes during and post the COVID pandemic period using the Arabic version of the DSMQ. It is interesting that the authors looked at changes in behaviors and self-management practices pre and post COVID. Given how the pandemic led to a lot of challenges and lifestyle changes, it is helpful to see this data.

Comments:

·       Are there studies in other populations where similar questionnaires (not necessarily the same) have been administered?

·       How was sample size calculated?

·       How were the questionnaires administered? Were these online questionnaires?

·       Do the authors have any reasoning for why most of the population was over the age of 50? Were most participants diagnosed with diabetes at an older age?

·       Since both type 1 and type 2 diabetic patients were included, was any analysis done limited to either type 1 or type 2. It might be possible that some of the self-management strategies practiced might differ based on whether the diagnosis was type 1 or type 2.

·       It would be helpful if the distribution of participants with type 1 and type 2 diabetes were presented.

·       Was data collected on participant demographics? It would help to see some of that data in a table.

Comments on the Quality of English Language

Minor editing to English language needed.

Author Response

(The authors gave the same response as above.)

Round 2

Reviewer 2 Report

Comments and Suggestions for Authors As mentioned response rate is very low, doi: 10.1007/s12029-021-00752-5. can be useful to improve the introduction.   Figure 1 is not needed, and is better in the text or table.   Is Table 2  needed and does it provide useful information?   I do not understand the reason for presenting table 3, is it in line with the purpose of the study?   Also, the figure 2 is confusing.   The title of Table 5 needs to be corrected.

Author Response

Thank you for your comments. Please find our responses attached. 
